# Heat Stress Tolerance: A Prerequisite for the Selection of Drought- and Low Phosphorus-Tolerant Common Beans for Equatorial Tropical Regions Such as Ghana

**DOI:** 10.3390/plants11182352

**Published:** 2022-09-09

**Authors:** David Appiah-Kubi, James Yaw Asibuo, Louis Butare, Stephen Yeboah, Zippora Appiah-Kubi, Alexander Wireko Kena, Henry Oppong Tuffour, Richard Akromah

**Affiliations:** 1Council for Scientific and Industrial Research, Crops Research Institute, Kumasi P.O. Box 3785, Ghana; 2Alliance of Bioversity International and International Center for Tropical Agriculture (CIAT), Cotonou 08 B.P. 0932, Benin or; 3Department of Crop and Soil Sciences, Faculty of Agriculture, Kwame Nkrumah University of Science and Technology (KNUST), Kumasi P.O. Box 152, Ghana

**Keywords:** climate change, food security, flower abortion, root characteristics, high temperature, drought tolerance, low phosphorus, common bean

## Abstract

Forty common bean accessions of multiple genetic background trait attribution regarding drought tolerance were selected based on mean yield performance from an earlier field test evaluation conducted using augmented RCBD. The various bean genotypes were further evaluated with phosphorus and water treatment interactions at two different levels for each factor. The experiment was conducted in a 2 × 2 × 40 factorial using RCBD with three replications under screen-house conditions at the CSIR-Crops Research Institute, Kumasi-Ghana. The objective was to select drought- and low phosphorus-tolerant common bean genotypes; which are suitable for tropical climatic conditions. The results showed that common bean with drought and heat trait tolerance survived, developed flowers and podded with seeds to physiological maturity, whilst genotypes with no heat trait tolerance had impaired reproductive structural development and growth disruption; thus, flowers could not develop into pods with seeds. This reproductive developmental anomaly was due to prevailing average daytime and nighttime high temperatures of 35.45 °C and 29.95 °C, respectively, recorded during the growth period, which reduced pollen fertility. Among the 478 experimental bean plants (two plants were missing) analyzed, 141 (29.5%) did not flower, 168 (35.18%) had their pods dropped whilst 99 (20.7%) podded with seeds to achieve physiological maturity. The podded-seed bean genotypes were of the SEF-line pedigrees, which were shown to be heat and drought-tolerant. Meanwhile, bean accessions with SMC, SMN and SMR code prefixes did not pod into seed despite possessing drought-tolerant traits. The effects of interactions between phosphorus and water treatments on the root characteristics of drought-tolerant common bean were as follows: root length, root surface area, average root diameter and root volume growth extensions doubled dimensionally under optimum conditions (P_2_W_2_) compared to stressed conditions (P_1_W_1_). The results from the present study identified four SEF-bean genotypes, namely, SEF15, SEF 47, SEF 60 and SEF 62, as superior yield performers, even under low soil phosphorus and in extreme high temperature conditions. Therefore, breeding for the selection of drought- and low-P-tolerant common bean for tropical agro-ecological environments must also consider concomitant heat stress tolerance.

## 1. Introduction

Common bean is nutrient-dense food for human consumption. It is a valuable source of essential amino acids such as lysine, vitamins such as niacin and thiamin, and minerals such as iron, calcium, zinc, phosphorus and magnesium [1,2]. Biotic and abiotic stress factors limit the development and production of common bean. Among abiotic stress factors, drought and low soil availability of phosphorus are critical constraints in bean production in tropical regions [2,3]. Drought stress occurs in plants because of a water deficit in the soil, which disrupts plant growth and development processes, thus affecting yield. The effect of drought on beans depends on their growth stage. Drought at flowering may cause flower drop and low pod set, while podding, it results in partial filling or shriveled seed formation. In either case, both quality and yield are reduced. 

In screening and selecting common bean lines for drought tolerance, it is important to strategically impose terminal drought at the reproductive growth stage, i.e., when plants are known to be sensitive to moisture stress [4,5].

In attempting to identify drought-tolerant bean for cultivation in the tropics, one underlying factor to consider is the heat tolerance level of the genotype. The common bean plant is sensitive to temperature, especially during reproduction phase. Its sensitivity to nighttime temperatures an even greater determining factor, as viable seeds are produced only when night temperatures are cool [6,7]. Heat stress causes a significant reduction in yield and quality. It has been reported that daytime temperatures above 30 °C and nighttime temperatures above 20 °C lead to a reduction in grain yield [7,8,9,10]. The major effects of high temperature include inhibition of pollen-fertility, flower drop and pod abortion [7,11,12]. The development of drought- and heat-tolerant bean varieties would also increase resilience to the food insecurity which will result from future global climate change. Global warming will have a negative impact on food production, especially in developing, tropical countries such as Ghana, where daily temperatures in the northern parts (the “food basket” of the country) could reach 40 °C during peak periods [13,14]. 

The co-occurrence of abiotic stresses such as drought, heat stress and low-P in soil has shown to be far more devastating for common bean production than the damage resulting from these phenomena separately. Drought and heat stress together could result in a 60% reduction in common bean production worldwide [15,16,17]. Breeding programs to improve the selection of common bean lines which are resilient to these three forms of abiotic stress could enhance yield and increase the land area which is suitable for cultivation. Bean genotypes tolerant to drought, low-P and heat stress have been identified [10,18] and developed from interspecific crosses of common bean (*Phaseolus*
*vulgaris*) with other *Phaseolus* species (*P*. *acutifolius*, *P*. *coccineus* and *P. dumosus*). Suarez (2020) used 92 bean genotypes, similar to the breeding lines used in this study, among which were: 23 “SEF” prefixed lines, developed from interspecific lines of *Phaseolus*
*vulgaris*, *P*. *acutifolius* and *P*. *coccineus*; 7 “SMC”-prefixed breeding lines from interspecific lines of *Phaseolus*
*vulgaris*. *P*. *acutifolius* and *P*. *dumosus*; and 8 “SMR” and 2 “SMN” prefixed advanced bio-fortified lines. The BFS lines are drought-tolerant and known to have enhanced adaptation to low soil fertility. The “SEF” and “SER” lines were developed to adapt to drought and heat tolerance. The SMC, SMR and SMN lines are tolerant to drought with high mineral (Fe) contents in seed [10]. The objective of this study was to identify common bean genotypes for tropical climatic conditions that are drought- and low-P-tolerant.

## 2. Results

### 2.1. Ambient Temperature and Humidity Conditions at the Screen-House

The maximum and minimum temperatures in the screen-house used in this study were 46.72 and 18.78 °C respectively. The maximum and minimum relative humidity were 97.9 and 17.75% (Table 1). N (Table 1) is the duration of the bean growth period, determined using a temperature/humidity data logger set at three-hour interval readings. Thus, the N = 391 multiplied by three gives an indication of the estimated number of hours (1173 h) required to achieve physiological maturity (49 days). 

The average daytime and nighttime temperatures recorded in the screen-house during the growth period of common bean were 35.45 °C and 24.95 °C, respectively (Table 2), and again, the average daytime and nighttime relative humidity conditions were 50.78% and 84.84%, respectively (Table 3).

A summary of the prevailing temperature and humidity conditions for the entire growth period at the screen-house is provided in Table 1.

The temperature readings were negatively correlated with relative humidity with a correlation coefficient value of −0.94 at a high level of significance (*p* = 0.001) (Table 4). 

### 2.2. Assessment of Reproductive Growth on Drought-Tolerant Common Bean Accessions in a Phosphorus–Water Factorial Treatment under Terminal Drought, Screen-House Conditions

Observations of the growth of various bean genotype accessions revealed differential response in terms of reproductive development; this was probably due to the prevailing high temperature conditions recorded in the screen-house. Among the 478 experimental bean plant-accessions, 141 could not flower (29.5%), 70 initiated flower buds but underwent flower-abortion (16.64%), 168 initiated pods but later dropped them (35.18%), and 99 achieved physiological maturity with seeds (20.72%), as shown in Table 5.

The number of common bean genotypes that could not flower was 141, which constituted 29.5% of the total experimental plants. Upon disaggregating phosphorus and water treatment interactions, it was found that non-flowering (no flower) common beans under optimal conditions that is, multiple non-stressed (P_2_W_2_) constituted 10.46% of the total, moisture stressed and low-P that is multiple stressed (P_1_W_1_) represented 4.18%. Low-P and well-watered (P_1_W_2_) treatment interaction comprised 3.56% whilst optimum-P and moisture stressed (P_2_W_1_) made up 11.3%. The percentage disaggregations of other character-trait observations concerning the four moisture–phosphorus treatment interactions are presented in Table 5.

As per the experimental design employed, each bean genotype had four-treatment combinations of water and phosphorus (P_1_W_1_, P1W2_,_ P_2_W_1_ and P_2_W_2_) per replication. Therefore, for the three replications in the design (RCBD), the maximum number for the ‘frequency of occurrence’ parameter for a genotype was 12 (i.e., 2.51% of the maximum). 

The majority of the SMC, SMN and SMR- prefixed bean accessions did not bear flowers (not even flower buds). For example, out of the 12 maximum frequency occurrences of experimental bean plants, seven, eight and nine plants of genotypes SMN 157, SMN 158 and SMN 159, respectively, did not flower (Table 6). Similarly, 10 and 11 of bean genotypes SMR 113 and SMR 118 did not flower. Finally, 9 out of the 12 experimental ‘Ennepa’ variety plants (non-drought check variety) failed to flower.

The SMC, SMN and SMR bean accessions again experienced high flower abortion rates. For genotypes BFS 35 and BFS 39, four and five plants, respectively, aborted their flowers. Similarly, three of the ‘Ennepa’ (non-drought check variety) plants aborted their flowers (Table 7).

Pod abortion predominantly seen in BFS and SEF prefixed-bean genotypes. All 12 BFS 67 genotype plants and 11 out of the 12 SEF 53 plants aborted their pods. Nine experimental plants each for the BFS 60 and SEF 73 genotypes aborted their pods (Table 8).

The SEF-bean accessions were most able to pod under the stress conditions applied in the screen-house. For instance, a total of 9, 12, 10 (SEF 47), 10 (SEF 60) and 11 plants of bean lines SEF 10, SEF 15, SEF 47, SEF 60 and SEF 62, respectively, achieved pods with seeds (Table 9).

### 2.3. Assessment of the Root Characteristics of Drought-Tolerant Common Bean Accessions in a Phosphorus–Water Factorial Treatment under Screen-House Conditions of Terminal Drought

The root lengths of various bean genotypes ranged from 5.5 and 10.9 m at low-P and moisture stress (P_1_W_1_) and optimum-P and well-watered (P_2_W_2_), respectively (Table 10).

The effects of phosphorus level and moisture factorial treatment on root characteristics (i.e., total root length, surface area, dimeter and volume) were measured using WinRhizo on drought-tolerant common beans under screen-house conditions (Table 10, Table 11, Table 12 and Table 13).

The mean root surface areas ranged from 410 to 907 cm^2^ for low-P/moisture stress and optimum-P/well-watered treatments, respectively (Table 11). The average root diameters for stress and non-stress conditions ranged from 0.82 to 1.64 mm, respectively (Table 12). The mean root volume also followed a similar trend for stress and non-stress conditions (Table 13). 

The response of seed-podded bean genotypes to phosphorus–water treatment combinations regarding root growth characteristics were as follows: genotypes AWASH 1, SEF28, SEF 55 and SMC 159 had mean root lengths of 10.359 cm, 10.929 cm, 10.203 cm, and 10.307 cm, respectively. The combined mean length of all bean genotypes was 7986 cm (Table 14). Bean genotypes with high podding frequency were SEF 15, SEF 47, SEF 60 and SEF 62, which had total root lengths of 5518 cm, 4833 cm, 3013 cm, 6468 cm, respectively. These values were lower than combined mean value of 7986 cm. 

### 2.4. Soil Moisture Content during Terminal Drought

At day-1 of terminal drought (i.e., one day after soil water saturation/satiation), the soil moisture content was 0.24 and 0.25 m^3^/m^3^ for P_1_W_1_ and P_2_W_1_ moisture stress treatments, respectively (Table 15). At day-5 of terminal drought, the soil moisture content of potted soil began to yield negative values, i.e., −0.01 m^3^/m^3^ for both moisture stress treatments (P_1_W_1_ and P_2_W_1_). At day-14, both moisture stress treatments read −0.09 m^3^/m^3^. 

## 3. Discussion

Drought stress during the reproductive stage is a major problem for common bean production because it affects the flowering and pod filling processes, thereby reducing yields [19,20]. Prior to our experiments, it was not known how prevailing ambient temperature and humidity conditions in a screen-house would selectively affect the reproductive phase of bean genotypes and the development of pods with seeds. High temperature as an abiotic stressor was among the factors considered; however, this became necessary over the course of this study. Despite the fact that the experiments were conducted in a confined, sheltered screen-house, the timing of the simulation of terminal drought stress was critical; it had to coincide with the dry season, as this is the most suitable period for the assessment and selection of promising drought-tolerant lines for local adaptation. In addition, adequate phenotypic and genotyping assessments facilitated the screening and selection of suitable drought-tolerant bean lines [4].

Among the 478 experimental plant units (there were two missing plants), only 20.7% were able to pod to maturity with seeds. It became apparent that the genotypes classified as drought- and heat-tolerant (based on available genetic background information) were able to pod with seeds. It is worth mentioning that neither optimal phosphorus and well-watered treatment interactions nor low phosphorus and moisture-stress treatment interactions translated into higher rates of pod formation and seed development. The common bean plant seems to have a sensitive physiological balance between the vegetative and reproductive phases, especially under drought, heat and low-P environmental conditions. As such, any of these abiotic factors could disrupt its growth and development. 

High temperature is concomitant to drought stress under tropical climatic conditions [21]. It is therefore critical that it is not overlooked when selecting drought-tolerant common bean varieties for tropical environments. It was revealing that the bean genotypes with drought- and heat-tolerance attributes were able to pod with seeds, whilst those without such traits, despite being tolerant to drought, could not pod into seed under prevailing screen-house conditions. 

Day and night temperatures play a crucial role in pod and seed development in the common bean [6]. The average daytime temperature of 35.45 °C and average nighttime temperature of 24.95 °C recorded during the study in the screen-house were above the ideal temperature conditions for the growth of common bean. The ambient temperature conditions in screen-house even reached a maximum of 46.72 °C. It has been established that average daytime temperatures above 30 °C and average nighttime temperatures above 20 °C reduce number of florets, increase the rate of flower abortion and subsequently reduce yield by 60% in common beans [10,22]. In our study, high temperatures truncated the growth of flower buds in some bean accessions while others had their flowers aborted and pods dropped, thereby reducing grain production [7,12].

In screening for heat-tolerant common beans under greenhouse conditions, Rainey and Griffiths (2005) [6] recommended a daytime temperature of 30 °C and nighttime temperature of 27 °C, for optimum yield. The bean accessions showed differential responses in the reproductive growth phases under high-temperature, screen-house conditions; i.e., accessions could seemingly be categorized, at their reproductive development stage, into four groups: not flowered (i.e., no flower bud), flower abortion, pod abortion and podded seed. The underlying mechanism for this was the presence of heat-tolerance traits in the genetic backgrounds of some of the studied bean accessions. Even though all the studied bean accessions possessed drought-tolerance attributes, only the genotypes with heat tolerance attributes as an additional trait achieved pod to maturity with seeds. As per the design and factorial arrangement with levels of treatment in the screen-house, the expected frequency for a given bean genotype was 12 (2.5%). None of the experimental plants with accession codes SMN and SMR matured to bear pod with seeds, even though they are drought tolerant. Therefore, in screening and selecting drought-tolerant common bean lineages for equatorial, tropical environments, heat stress tolerance is a pre-requisite.

If excessively warm temperatures can prevent the reproductive structures of common bean plants from developing into seeds, then it stands to reason that the effects of climate change will have serious consequences on food security. It has been predicted that global temperatures will rise by 2.0 °C by 2050 as result of climate change [23,24]. Therefore, the creation of climate-resilient crop genotypes requires urgent attention, especially in developing countries in sub-Saharan Africa. 

Our results identified four SEF-bean lines, namely, SEF 15, SEF 47, SEF 60 and SEF 62 as superior in terms of heat tolerance, as depicted in their high ‘frequency occurrence’ of podded seed formation under low-P and drought conditions. Beebe et al. [25] reported similar results, i.e., five SEF bean lines (SEF 14, SEF 15, SEF 16, SEF 43 and SEF 60) stood out as superior in terms of their heat tolerance and grain yield. In the present study, four SEF genotypes (SEF 15, SEF 47, SEF 60 and SEF 62) were consistent in terms of superior yield under both low-P and drought stress (P_1_W_1_) and optimum-P and well-watered (P_2_W_2_) conditions. With respect to the Ennepa (non-drought) variety, growth was truncated at the flower-abortion stage (see Table 10 and Table 11). Common bean accessions with codes SMC, SMN and SMR, despite being drought-tolerant, were not able to cope with the heat stress they experienced in the screen-house. This resulted in low pollen growth viability and, thus, affected grain yield. The SEF-bean lines were drought-tolerant and better adapted to heat stress, providing superior grain yields. These results are in line with those of Suarez et al. [10].

The effects of treatment interactions of phosphorus and water combinations mainly manifested in the root growth characteristics of the studied bean genotypes. The root length, surface area, diameter and volume almost doubled in non-stress (P_2_W_2_) samples. For instance, a root length of 5453 cm was observed under stressed condition (P_1_W_1_) while one of 10,882 cm was measured under non-stressed condition (P_2_W_2_). Similarly, a root surface area of 410 cm^2^ was observed under stressed condition (P_1_W_1_) compared to 907 cm^2^ under non-stress condition (P_2_W_2_). The same trend was observed for root diameter and volume (see Table 10, Table 11, Table 12 and Table 13). Phosphorus treatment is known to drive root growth in common bean. The synergistic effect of phosphorus and water on growth among drought-tolerant common beans was more evident in the root characteristics than in the yield. The root diameters in drought-tolerant genotypes appeared to have an inverse relationship with the ‘frequency of occurrence of pods’ parameter for a podded genotype. For instance, bean genotypes with superior grain yields (that is, higher ‘frequency of occurrence of pods’) had smaller root diameters (<1.0 mm), while genotypes with bigger root diameters (>1.0) had low grain yields (lower ‘frequency of occurrence of pods’) (Table 9 and Table 14). The measurements of root length, surface area, average diameter and volume were consistent and with R-square index values (index of strength) ranging from 0.73–0.76. The results of the root analysis based on WinRhizo data revealed a CV range of 39–47%; this might have been due to the extremes of factorial treatment combinations, i.e., stressed (P_1_W_1_) versus non-stressed (P_2_W_2_) due to phosphorus and water application. Results obtained using WinRhizo are known to be comparable to manual measurements [26,27]. 

In terms of the root growth characteristics, there appeared to be a ‘redundant-effect’ of phosphorus application on drought-tolerant common bean, as both drought- and low-P-tolerance common beans had the tendency to produce extensive root systems. This is in line with findings by other authors [28,29]. For example, the mean root length of podded bean genotypes ranged from 5453 cm for P_1_W_1_ to 10,882 cm for P_2_W_2_, and mean root surface areas ranged from 410 cm^2^ for P_1_W_1_ to 907 cm^2^ for P_2_W_2_ (Table 10, Table 11, Table 12 and Table 13). Even under the stressed-treatment condition (P_1_W_1_), appreciably high root growth measurements were recorded. The results from this study therefore showed that drought- and low-P-tolerant common beans were able to adapt in terms of their root growth characteristics. Similar results and observations have been reported, with common bean plants showing enhanced root-to-shoot ratios for increased yield in P-deficient soils [30,31]. Root depth in common bean is important component in determining overall drought tolerance. The application of phosphorus to drought-tolerant common beans further increased root growth.

When moisture content readings were positive, the soil aggregates contained surface moisture; however, if the moisture content was negative, it implied that the aggregates were dry [32]. The negative soil moisture values recorded (−0.01 to −0.09 m^3^/m^3^) in the potted soil from day 5 to 14 of pod development indicated that the bean genotypes had endured and survived ten days of soil surface dryness. The effect of soil compaction was negligible, as there was no significant difference in mean bulk densities among potted soils (data not shown); this, otherwise, could have influenced the rate of water movement and root growth extension within the soil medium. 

The results of our examination of phosphorus–water factorial treatment interactions revealed that phosphorus application manifested more during the agronomic vegetative stage, whilst well-watered treatment was vital for the development of yield components such as pods. The response to phosphorus–water treatment combinations regarding the phenology (that is, days to flowering, days to 50% flowering and days of maturity) of the bean lines was influenced by genetic constitution rather than the treatments themselves [33]. This ‘constitutive trait’, as proposed by Chaves et al. [33], manifested in the bean genotypes with genetic backgrounds of drought and heat tolerance. Meanwhile, drought- and heat-sensitive genotypes did not pod into seed, even under optimal phosphorus–water conditions. The genotypic differences among drought-tolerant bean accessions were attributed to the differential responses of reproductive structures as a result of the ambient temperature conditions in the screen-house.

## 4. Materials and Methods

### 4.1. Screen-House Factorial Experiment of 40 Selected Common Bean Lines with Phosphorus and Water as Treatment Factors

The screen-house was a galvanized metal frame structure in which each side was covered with size-50 Aphid-proof mesh net to reduce external influence and to ensure a confined environment. The roof metal frame comprised a dome-shaped arch covered with transparent rain-off shelter made of poly sheet. The screen-house had a dimensional space of 15.0 m floor length, 10.0 m arc-length, 8.0 m floor width and arch heights of 4.5 m at the middle lane and 1.7 m at the outer lane (edges). The structure was located at the CSIR-Crops Research Institute, Fumesua-Kumasi, Ghana. The experiment was laid out as a 2 × 2 × 40 factorial in a randomized complete block design (RCBD) with three replications. There were three treatment factors namely: common bean genotype, phosphorus and water. The phosphorus and water treatment factors were assessed at two levels each, i.e., low/optimum phosphorus and low (moisture stress at terminal drought)/well-watered treatments. Overall, there were 160 treatment combinations, i.e., 2 levels of phosphorus application × 2 levels of water application × 40 bean genotypes. Each treatment combination was replicated three times, making a total of 480 experimental plant units. A list of 40 drought-tolerant common bean varieties was used in the experiment (Table 16). Experiments took place in February 2019, which is still within the dry season in Ghana. The dry season is usually characterized by low humidity and dry air with no rain. 

Two levels of phosphorous fertilizer were applied: an optimal application of 100 kg P/ha of triple super phosphate (_TSP_) equivalent (P_2_), i.e., 0.112 g of TSP/kg soil; and a low-P soil (P_1_), i.e., unamended soil. The total amount of optimum TSP used per pot was 0.364 g (see Appendix A).

Two levels of water regimes were studied: well-watered (equal volumes of water to be measured) from germination to maturity, followed by low-watered up to the vegetative stage, and finally, terminal drought (withdrawal of water) during the podding stage (i.e., the appearance of a first pod of about 3 cm in length); and well-watered at podding as a control. These factors formed the basis of the factorial experimental treatment interactions (T_1_, T_2_, T_3_, and T_4_), as illustrated in Table 17.

The effect of interactions among these parameters on crop performance were assessed on bean genotypes by agronomic, phenology and yield components and other root characteristics.

### 4.2. Soil Sample Collection and Laboratory Analysis

A demarcated land area (20 × 20 m) (known or suspected to be low in soil phosphorus) was cleared of weeds, and soil was collected from a depth of 0.0–20.0 cm and steam sterilized at 121 °C for 30 min before chemical analysis. Then, following a chemical assessing of the P level, a 3.25-kg sterilized soil sample (loamy sand) was placed in each column pot. The column-pots used in this study comprised PVC pipe cut to a length of 24.2 cm with a fixed diameter of 11.3 cm, vertically seated in a perforated-cover at the bottom to allow excess water to drain off.

The results of the laboratory chemical and physical analyses of the soil sample are shown in Table 18 and Table 19, respectively. Soil samples were analyzed chemically using the Bray-1 method to assess the level of available phosphorus, providing a baseline for the amount of triple superphosphate (TSP) to be applied for phosphorus treatment. The available P in the sample was low (5.4 mg/kg soil) justifying its use for the low-P (P-stress) condition experiments. The pH was measured at a soil:water ratio of 1:2.5. The organic carbon content was estimated by the wet oxidation Walkley/Black method, and total nitrogen was estimated by the macro-Kjeldahl method.

### 4.3. Temperature Data Logger at the Screen-House

A temperature–humidity data logger sensor (Supco^®^. SL500TH. SN: 05151131CF, Allenwood, NJ, USA) was activated to record and save temperatures and humidity values at three-hour intervals for the growing period of common bean in the screen-house. A representative graph of temperature/humidity/dew is provided in Figure 1.

### 4.4. Moisture-Pro-Check 10HS Soil Moisture Sensor (Decagon Devices, Inc.)

We used a Moisture-Pro-Check 10HS Soil Moisture Sensor to determine the soil moisture and water contents as m^3^/m^3^ (volumetric water content, i.e., 1 m^3^ water/1 m^3^ total soil volume). Soil moisture content is defined as the volume of water, V*w*, that is removed from volume of soil V*s* by drying the soil at 105 °C. A meter probe (sensor) was placed inside the potted soil and readings were recorded. Subsequent measurements were taken from same hole in order to avoid excessive soil disturbance.

### 4.5. Imposing Terminal Drought

Potted soil (3.25 kg soil sample) was satiated/saturated with 750 mL water in the evening following pod initiation stage. Excess water was drained until the following morning (12 h) under gravity. Soil moisture measurements were taken with a ProCheck-10HS moisture sensor (DECAGON DEVICES) the following morning and on all subsequent mornings up to Day 14. No further water was applied in order to achieve terminal drought conditions. Soil moisture measurements (m^3^/m^3^) were taken every morning during the 14-day terminal drought period.

### 4.6. Root Characteristics Measurement Using WinRHIZO™

WinRHIZO is an image analysis software specifically designed for automatic root measurements via a digitized scanner. Shoot portions were obtained at soil level. Harvested roots then removed from PVC columns (pots) and washed thoroughly in a pan of water to remove soil particles. Labelled, washed-roots were kept in plastic bag in a refrigerator until measurements of root characteristics had been completed. Washed roots were spread out on a scanner (avoiding overlap of secondary roots) for image acquisition. As root growth mass was extensive, as precautionary measure, it was cut into pieces for measurements, after which the sum values were recorded. It was a laborious work to prepare the roots for the WinRhizo measurements; therefore, only bean genotypes that had undergone moisture stress (P_1_W_1_ and P_2_W_1_) and which were podded with seeds were assessed. Root length, surface area, diameter and volume were recorded. After scanning, the individual root parts were gathered for oven drying. 

### 4.7. Statistical Analyses

Statistical analyses were conducted using SAS (Statistics, version 9.4. SAS Institute, Cary, NC, USA). A general linear model (GLM) was used for data analysis. Analyses of variance (ANOVA) for each measured variable were performed using the generalized linear model (GLM). Pearson’s correlation analyses were conducted on variable traits all at levels of significance at alpha = 0.05.

## 5. Conclusions

Breeding for the selection of drought- and low-P-tolerant common bean for tropical agro-ecological environments must also consider concomitant heat stress tolerance. The present study revealed how some drought-tolerant common bean accessions are susceptible to high temperature, resulting in growth disruption and termination of reproductive structural development, thereby affecting yield output. The study also showed moderate root growth characteristics among drought- and low-P-tolerant common beans. Global warming is a threat to food security, especially in tropical settings. Therefore, to ensure sustained global food security, urgent research efforts are needed to breed resilient crops with multiple adaptive traits allowing them to thrive under emerging climate change conditions.

## Figures and Tables

**Figure 1 plants-11-02352-f001:**
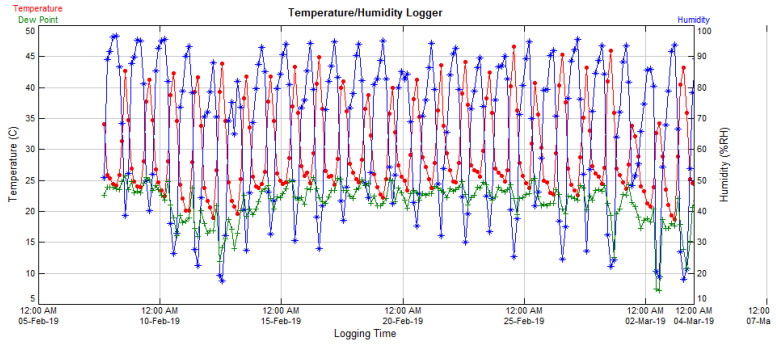
A graph of temperature-humidity-dew conditions from the vegetative growth to the post-flowering period of common bean in a screen-house, obtained using a Supco^®^ data logger sensor.

**Table 1 plants-11-02352-t001:** Statistics of temperature and humidity conditions at the screen-house.

Parameter	N	Mean	StdErr	StdDev	Min	Max
Temperature (°C)	391	30.18	0.35	6.83	18.78	46.72
Rel. Humidity (%)	391	67.86	1.14	22.55	17.75	97.9
Dew Point (°C)	391	22.28	0.13	2.48	7.23	25.94

N = 3 h interval temperature data point from the period of emergence through flowering to physiological maturity; Min = minimum; Max = maximum; StdErr = standard error of mean; StdDev = standard deviation of mean.

**Table 2 plants-11-02352-t002:** Day and night temperature conditions at the screen-house.

Variable	Nighttime Temperature (°C)	Daytime Temperature (°C)
Avg. Temp	24.95	35.45
Max. Temp	30.72	46.72
Min. Temp	18.78	23.08

**Table 3 plants-11-02352-t003:** Day and night relative humidity conditions at the screen-house.

Variable	Nighttime % Rel. Humidity	Daytime % Rel. Humidity
Avg. Rel. Humidity	84.84	50.78
Max. Rel. Humidity	97.9	90.45
Min. Rel. Humidity	53.95	17.75

**Table 4 plants-11-02352-t004:** Pearson’s correlation coefficients for temperature, relative humidity and dew for N = 391 data points.

	Temp (°C)	Rel. Humidity (%)	Dew Point (°C)
Temp (°C)	1		
Rel. Humidity (%)	−0.935 ***	1	
Dew Point (°C)	0.095 ns	0.214 ***	1

*** = Significant at 0.001; ns = not significant at 0.05.

**Table 5 plants-11-02352-t005:** Flowering and pod formation characteristics of 478 drought-tolerant common bean plants in factorial experiments with phosphorus–water treatments under sheltered-roof screen-house conditions.

Reproductive Character Status on Bean Genotypes	Phosphorus–Water Factorial Treatments	Frequency Occurrence	Percentage Occurrence
No Flower	P_1_W_1_	20	4.18
No Flower	P_1_W_2_	17	3.56
No Flower	P_2_W_1_	54	11.3
No Flower	P_2_W_2_	50	10.46
Total Not Flowered		141	29.5
Flower Aborted	P_1_W_1_	15	3.14
Flower Aborted	P_1_W_2_	16	3.35
Flower Aborted	P_2_W_1_	15	3.14
Flower Aborted	P_2_W_2_	24	5.02
Total Flower Aborted		70	14.65
Pod Aborted	P_1_W_1_	59	12.34
Pod Aborted	P_1_W_2_	54	11.3
Pod Aborted	P_2_W_1_	31	6.49
Pod Aborted	P_2_W_2_	24	5.02
Total Pod Aborted		168	35.15
Pod Matured	P_1_W_1_	26	5.44
Pod Matured	P_1_W_2_	33	6.9
Pod Matured	P_2_W_1_	20	4.18
Pod Matured	P_2_W_2_	20	4.18
Total Pod Matured		99	20.7

SAS General Linear Model (GLM) ANOVA at alpha = 0.05; Treatment Code: P_1_W_1_ = low P + low-water; P_1_W_2_ = low P + well-watered; P_2_W_1_ = optimum P + low water; P_2_W_2_ = optimum P + well-watered.

**Table 6 plants-11-02352-t006:** Non-flowering frequency of drought-tolerant common bean genotypes during terminal drought.

Genotype	Frequency Occurrence	Percent Occurrence	Cumulative Frequency	Cumulative Percentage
AWASH1	7	1.46	7	1.46
BFS30	4	0.84	11	2.3
BFS35	5	1.05	16	3.35
BFS39	1	0.21	17	3.56
BFS55	2	0.42	19	3.98
BFS59	6	1.26	25	5.24
BFS60	1	0.21	26	5.45
BFS62	1	0.21	27	5.66
ENNEPA (check)	9	1.88	36	7.54
SEF17	1	0.21	37	7.75
SEF60	1	0.21	38	7.96
SEF73	2	0.42	40	8.38
SMC146	5	1.05	45	9.43
SMC155	4	0.84	49	10.27
SMC157	7	1.46	56	11.73
SMC158	9	1.88	65	13.61
SMC159	8	1.67	73	15.28
SMC160	4	0.84	77	16.12
SMC161	5	1.05	82	17.17
SMN58	8	1.67	90	18.84
SMN63	8	1.67	98	20.51
SMR101	8	1.67	106	22.18
SMR103	2	0.42	108	22.6
SMR107	3	0.63	111	23.23
SMR113	11	2.3	122	25.53
SMR118	10	2.09	132	27.62
SMR127	5	1.05	137	28.67
SMR128	4	0.84	141	29.51

**Table 7 plants-11-02352-t007:** Flower abortion frequency of drought-tolerant common bean genotypes during terminal drought.

Genotype	Frequency Occurrence	Percent Occurrence	Cumulative Frequency	Cumulative Percentage
AWASH1	1	0.21	1	0.21
BFS30	2	0.42	3	0.63
BFS35	4	0.84	7	1.46
BFS39	5	1.05	12	2.51
BFS55	3	0.63	15	3.14
BFS59	1	0.21	16	3.35
BFS60	2	0.42	18	3.77
BFS62	3	0.63	21	4.39
ENNEPA (check)	3	0.63	24	5.02
SEF17	1	0.21	25	5.23
SEF28	2	0.42	27	5.65
SEF44	1	0.21	28	5.86
SEF53	1	0.21	29	6.07
SEF62	1	0.21	30	6.28
SEF73	1	0.21	31	6.49
SMC146	1	0.21	32	6.69
SMC155	3	0.63	35	7.32
SMC157	4	0.84	39	8.16
SMC158	3	0.63	42	8.79
SMC159	3	0.63	45	9.41
SMC160	4	0.84	49	10.25
SMC161	2	0.42	51	10.67
SMN58	4	0.84	55	11.51
SMN63	1	0.21	56	11.72
SMR101	2	0.42	58	12.13
SMR103	2	0.42	60	12.55
SMR107	2	0.42	62	12.97
SMR113	1	0.21	63	13.18
SMR118	2	0.42	65	13.6
SMR127	3	0.63	68	14.23
SMR128	2	0.42	70	14.64

**Table 8 plants-11-02352-t008:** Pod abortion frequency of drought-tolerant common bean accessions during terminal drought.

Genotype	Frequency Occurrence	Percent Occurrence	Cumulative Frequency	Cumulative Percentage
AWASH1	2	0.42	2	0.42
BFS30	6	1.26	8	1.68
BFS35	3	0.63	11	2.31
BFS39	6	1.26	17	3.57
BFS55	7	1.46	24	5.03
BFS59	3	0.63	27	5.66
BFS60	9	1.88	36	7.54
BFS62	5	1.05	41	8.59
BFS67	12	2.51	53	11.1
SEF10	3	0.63	56	11.73
SEF17	5	1.05	61	12.78
SEF28	7	1.46	68	14.24
SEF29	7	1.46	75	15.7
SEF44	6	1.26	81	16.96
SEF47	2	0.42	83	17.38
SEF52	7	1.46	90	18.84
SEF53	11	2.3	101	21.14
SEF55	4	0.84	105	21.98
SEF60	1	0.21	106	22.19
SEF64	3	0.63	109	22.82
SEF73	9	1.88	118	24.7
SMC146	5	1.05	123	25.75
SMC155	5	1.05	128	26.8
SMC157	1	0.21	129	27.01
SMC160	4	0.84	133	27.85
SMC161	5	1.05	138	28.9
SMN63	3	0.63	141	29.53
SMR101	2	0.42	143	29.95
SMR103	8	1.67	151	31.62
SMR107	7	1.46	158	33.08
SMR127	4	0.84	162	33.92
SMR128	6	1.26	168	35.18

**Table 9 plants-11-02352-t009:** Seed-podding frequency of drought-tolerant bean genotypes during terminal drought under screen-house conditions.

Genotype	Frequency Occurrence	Percent Occurrence	Cumulative Frequency	Cumulative Percentage
AWASH1	2	0.42	2	0.42
BFS59	2	0.42	4	0.84
BFS62	3	0.63	7	1.47
SEF10	9	1.88	16	3.35
SEF15	12	2.51	28	5.86
SEF17	4	0.84	32	6.7
SEF28	2	0.42	34	7.12
SEF29	5	1.05	39	8.17
SEF44	5	1.05	44	9.22
SEF47	10	2.09	54	11.31
SEF52	5	1.05	59	12.36
SEF55	8	1.67	67	14.03
SEF60	10	2.09	77	16.12
SEF62	11	2.3	88	18.42
SEF64	9	1.88	97	20.3
SMC146	1	0.21	98	20.51
SMC159	1	0.21	99	20.72

**Table 10 plants-11-02352-t010:** Effect of phosphorus–water treatments on root-length of common bean.

Treatment Levels	N	Mean Total Root Length (cm)	Mean StdErr	Min. Length	Max. Length
P_1_W_1_	34	5453.82	692.33	488.41	17,209.56
P_1_W_1_	35	7605.21	540.57	2221.37	14,141.84
P_2_W_1_	32	8017.05	907.61	142.69	19,636.44
P_2_W_2_	34	10,882.05	578.54	5413.16	17,714.55

**Table 11 plants-11-02352-t011:** Effects phosphorus–water treatments on root surface area (cm^2^) of common bean.

Treatment Levels	N	Mean Root Surface Area (cm^2^)	Mean StdErr	Min. Surface Area	Max. Surface Area
P_1_W_1_	34	410.58	55.24	31.46	1299.13
P_1_W_2_	35	600.37	44.42	147.99	1238.19
P_2_W_1_	32	650.18	77.74	12.98	1692.43
P_2_W_2_	34	907.17	49.73	449.36	1468.25

**Table 12 plants-11-02352-t012:** Effects of phosphorus–water treatment on root diameter (mm) of common bean.

Treatment Levels	N	Mean Root Diameter (mm)	Mean StdErr	Min. Diameter	Max. Diameter
P_1_W_1_	34	0.82	0.1	0.19	2.07
P_1_W_2_	35	1.19	0.08	0.43	2.5
P_2_W_1_	32	1.19	0.13	0.21	2.76
P_2_W_2_	34	1.64	0.1	0.75	3.08

**Table 13 plants-11-02352-t013:** Effects of phosphorus–water treatments on root volume (cm^3^) of common bean.

Treatment Levels	N	Mean Root Volume (cm^3^)	Mean StdErr	Min. Volume	Max. Volume
P_1_W_1_	34	2.51	0.36	0.16	7.83
P_1_W_2_	35	3.81	0.3	0.79	8.64
P_2_W_1_	32	4.24	0.54	0.09	11.62
P_2_W_2_	34	6.08	0.36	2.9	10.27

**Table 14 plants-11-02352-t014:** Response of root characteristics of seed-podded, drought-tolerant common bean genotypes measured using WinRhizo under phosphorus and moisture treatments during terminal drought.

Common Bean Genotype	Mean Root Length (cm)	Mean Surface Area (cm^2^)	Average Root Diameter (mm)	Mean Root Volume (cm^3^)
AWASH1	10,359.04	825.64	1.34	5.27
BFS59	9466.82	744.26	1.41	4.72
BFS62	7880.31	652.21	1.26	4.39
SEF10	9630.07	772.49	1.41	5.00
SEF15	5518.09	424.08	0.83	2.62
SEF17	8084.45	650.05	1.32	4.22
SEF28	10,929.92	879.14	1.58	5.68
SEF29	9923.6	803.75	1.59	5.23
SEF44	6749.95	550.07	0.97	3.61
SEF47	4833.63	382.98	0.79	2.44
SEF52	9654.93	825.69	1.6	5.66
SEF55	10,203.17	813.87	1.43	5.24
SEF60	3013.99	224.02	0.47	1.34
SEF62	6468.47	489.13	0.92	2.97
SEF64	7983.79	610.03	1.11	3.74
SEF73	6272.05	489	0.97	3.08
SMC146	5112.89	435	0.97	3.02
SMC159	10,307.57	872.81	1.64	5.93
CV %	39.46	42.82	43.92	47.7
Mean	7986.28	641.64	1.21	4.16
MSE	3151.16	274.76	0.53	1.98
R-square	0.76	0.75	0.69	0.73

Coefficient of variation of all root trait measurements, measured using a WinRhizo analyzer. Values ranged from 39–47% with R-square index values also ranging from 0.73–0.76 (Table 14).

**Table 15 plants-11-02352-t015:** Means (pooled) of soil moisture content (m^3^/m^3^) in potted soil of pod-seeded common bean plants during a 14-day terminal drought stress period under phosphorus–water treatment in a screen-house.

Days	P_1_W_1_ Soil Moisture (m^3^/m^3^)	P_2_W_1_ Soil Moisture (m^3^/m^3^)
1	0.25	0.24
2	0.17	0.13
3	0.1	0.07
4	0.02	0.01
5	−0.01	−0.01
6	−0.03	−0.06
7	−0.03	−0.06
8	−0.04	−0.07
9	−0.06	−0.08
10	−0.06	−0.09
11	−0.07	−0.09
12	−0.07	−0.09
13	−0.07	−0.09
14	−0.09	−0.09

P_1_W_1_ = low phosphorus and moisture stress; P_2_W_1_ = optimum phosphorus and moisture stress.

**Table 16 plants-11-02352-t016:** List of selected best 40 performing drought-tolerant common bean lines used in screen-house experiment.

1.SMR_113	11. SEF_29	21. SMC_160	31. SEF_62
2. SMC_158	12. BFS_35	22. SEF_64	32. SMC_161
3. SEF_17	13. BFS_60	23. BFS_59	33. SMC_146
4. SMN_63	14. SMR_107	24. SEF_28	34. SEF_47
5. SMN_58	15. BFS_39	25. SEF_55	35. BFS_67
6. BFS_30	16. SMC_155	26. SEF_10	36. ENNEPA (non-drought check)
7. SMC_159	17. SMR_103	27. SEF_52	37. SEF_15
8. SMR_118	18. SMR_101	28. BFS_62	38. SMR_129
9. SMC_157	19. AWASH_1	29. SMR_128	39. BFS_55
10. SMR_127	20. SEF_60	30. SEF_44	40. SEF_68

**Table 17 plants-11-02352-t017:** Illustration of the factorial experiment.

T_1_ = P_1_W_1_	T_2_ = P_1_W_2_	T_3_ = P_2_W_1_	T_4_ = P_2_W_2_

Key: P_1_ = Low Phosphorus; P_2_ = Optimum Phosphorus: W_1_ = Low-Watered; W_2_ = Well-Watered. T_1_–T_4_ = Respective Treatments.

**Table 18 plants-11-02352-t018:** Chemical analysis of soil used in the screen-house experiment.

Sample	pH	Available P (mg/kg)	% Total N	Exch. Bases (cmol/kg)				Exch. Acidity (cmol/kg)		% Org. Carbon	% Org. Matter
				K	Ca	Mg	Na	Al	H		
Soil Sample	6.55	5.445	0.0824	0.135	1.51	1.04	0.042	0.508	0.345	1.077	1.857

**Table 19 plants-11-02352-t019:** Physical analysis of soil used in the screen-house experiment.

Sample	% Sand	%Clay	% Silt	Textural Class	Water Holding Capacity
Soil Sample	83.20	6.40	10.40	Loamy sand	14

## Data Availability

Not applicable.

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
