# Peer review of "Heat Stress Tolerance: A Prerequisite for the Selection of Drought- and Low Phosphorus-Tolerant Common Beans for Equatorial Tropical Regions Such as Ghana"

_plants, 2022, doi:10.3390/plants11182352_

Round 1
Reviewer 1 Report
In this ms 40 common bean accessions in a genetic background of drought tolerance were subjected to terminal drought and low phosphorus treatment at the reproductive stage in a screen house. The experiment was a randomized complete block design with 3 factors – 1 - genotype, 2 - low and optimal soil P content (P1 vs P2), 3 – drought or optimal watering (W1 vs W2), with 3 replicates. Four factorial treatment interactions were distinguished – P1W1, P1W2, P2W1 and P2W2. Plant phenology and yield, as well as root morphology were registered along with records of the temperatures in the screen house. Actually, ambient temperature imposed additional heat stress on bean plants and not surprisingly, the majority of the tested accessions responded with no flowering/flower abortion/pod abortion. Only some SEF line pedigrees, which were interspecific lines combining Ph. Vulgaris, Ph. Acutifolius and Ph. Coccineus and had presented before heat tolerance along with drought tolerance, gave viable seeds. Root morphological traits were influenced by the treatments but did not show direct relation to yield.
Not many time-consuming, laborious experiments have been undertaken so far focusing especially on relevant stresses at the reproductive stage and in this respect the ms contain valuable new information. The conclusion is evident that breeding for heat tolerance is essential for common bean in the conditions of Ghana.
Remarks for ms improvement:
The English is understandable but should be checked for small mistakes.
More details about the screen house experiment should be given.
Lines 98-99 and 116-117 – repetition
Table 10a, 10b, 10c, 10d – column 3 – everywhere is Mean total root length – please correct
Line 233 – heat tolerant
Line 290 – non-drought variety – unclear. Did you mean drought non-tolerant variety?
Line 402 – soil?
Reviewer 2 Report
The manuscript by Appiah-Kubi et al. focuses on the characterization of the response to high temperatures and phosphorus availability of a set of Phaseolus genotypes previously classified as drought tolerant. The work arrives at a good characterization and classification of the genotypes without going into details of what may be the strategies that each one of them has to face with more or less luck the combination of abiotic factors studied. Beyond the quantification of reproductive parameters and root development, no further information is presented.
However, the work is well developed and is in line with the objectives set by the authors.
Comments:
Line 106. Please indicate the value of temperature more frequent (mode) both for minimum and maximum values. This parameter is important to analyse the environmental situation in the greenhouse.
Line 243-246. This affirmation should be supported by information or previous studies, please add references.
Line 307-308 Please be more precise with the idea of this sentence seems to be contradictory the following discussion is not clarifying.
Line 331-332 the idea expressed in this sentence, Did it came from another study? This study did not include the NDVI index for genotypes characterization.
M&M
Table 16. Indicate the meaning of 14 as the Water holding capacity index
Section 4.3 M&M. Because the temperature was monitored continually every 3 hours, authors should include a figure with all data set of temperature during all period of the assay.
